# The Effect of Melasma on the Quality of Life in People with Darker Skin Types Living in Durban, South Africa

**DOI:** 10.3390/ijerph20227068

**Published:** 2023-11-16

**Authors:** Nomakhosi Mpofana, Michael Paulse, Nceba Gqaleni, Mokgadi Ursula Makgobole, Pavitra Pillay, Ahmed Hussein, Ncoza Cordelia Dlova

**Affiliations:** 1Dermatology Department, Nelson R. Mandela School of Medicine, University of KwaZulu-Natal, Durban 4000, South Africa; dlovan@ukzn.ac.za; 2Department of Somatology, Durban University of Technology, Durban 4000, South Africa; mokgadim@dut.ac.za; 3Faculty of Health and Wellness Sciences, Cape Peninsula University of Technology, Cape Town 8000, South Africa; paulsemi@cput.ac.za; 4Discipline of Traditional Medicine, University of KwaZulu-Natal, Durban 4000, South Africa; gqalenin@ukzn.ac.za; 5Faculty of Health Sciences, Durban University of Technology, Durban 4000, South Africa; 6Department of Biomedical and Clinical Technology, Durban University of Technology, Durban 4000, South Africa; pillayp@dut.ac.za; 7Department of Chemistry, Cape Peninsula University of Technology, Cape Town 8000, South Africa; mohammedam@cput.ac.za

**Keywords:** melasma, pigmentation, darker skin type, Fitzpatrick skin types IV–VI, quality of life

## Abstract

Melasma is a common skin disorder of acquired hyperpigmentation that appears commonly on the face. Although asymptomatic, melasma causes psychosocial and emotional distress. This study aimed to assess melasma’s severity on people with darker skin types, evaluate the effects of melasma on the quality of life (QoL), and establish QoL predictors in affected individuals. This was a cross-sectional analytic study that enrolled 150 patients from three private dermatology clinics in Durban, South Africa who were diagnosed with melasma. The severity of melasma alongside QoL were measured using a melasma area and severity index (MASI) score and melasma quality of life scale (MELASQoL), respectively. The associations among factors and QoL were explored using multivariable methods and stepwise regression analysis. *p*-values less than 0.05 were considered significant. Enrolled patients were predominantly females (95%), of which 76% were of black African ethnicity, 9% were of Indian ethnicity, and 15% had mixed ancestry, with an average age of 47.30 years. Family history revealed that 61% had no prior melasma cases, while 39% had affected relatives, most commonly mothers (41%). The cheeks were the most common site for melasma. MASI score of Masi (β = 0.209, *t* = 2.628, *p* < 0.001), the involvement of cheeks (β = −0.268, *t* = −3.405, *p* < 0.001), level of education (β = −0.159, *t* = −2.029, *p* = 0.044), and being menopausal (β = −0.161, *t* = −2.027, *p* = 0.045) were found to be predictors of QoL. A regression model was created to forecast MELASQoL using these four predictors. This equation’s significance lies in its ability to enable the remote assessment of MELASQoL based on these four variables. It offers a valuable tool for researchers and medical professionals to quantitatively and objectively evaluate the impact of melasma on an individual’s quality of life.

## 1. Introduction

Skin illnesses are among the most frequent health problems in the world [1,2]. Superficially, this may seem less severe when compared to other health problems; however, the burden of skin disease is an intricate and complex concept that can encompass emotional and social interactions, as well as economic impacts on individuals, their families, and society [3]. Healthy skin is essential so that homeostasis can run normally and also to prevent various disorders and diseases [1,2]; therefore, the impact of skin conditions on well-being is proportional to their visibility [4]. The skin is the most extensive organ of the human body, and it serves the most important function as a protective organ from the environment [1,2]. The most common complaints dermatologists deal with are premature aging, acne, and pigmentary disorders, including melasma [5,6].

Melasma is a common dyschromia, mainly found in women between Fitzpatrick skin types IV–VI [7,8,9]. Melasma prevalence varies across the globe. Its prevalence as a multifactorial disorder has ranged from 1% in the population as a whole to 9–50% in populations at higher risk [10]. This broad variance in prevalence has been attributed primarily to differences in ethnicity and levels of sun exposure among population groups living in different geographic regions. The majority of these studies were conducted in general or dermatology clinics, which may not accurately reflect the prevalence in the studied region’s overall population; however, they suggest a high prevalence of melasma in the populations included in the studies. A study by Walker et al. conducted in 2008 suggests that in 546 dermatological patients in rural Nepal, melasma was the most common pigmentary disorder and the fourth most common dermatosis [11]. A 10-month retrospective study of common skin diseases in the Arab population in Saudi Arabia reported a prevalence of 7% of pigmentary disorders in 2017, whereas Arabs in America had a prevalence of 13.4–15.5% [12,13]. There was an 8.2% prevalence among 1000 Latino patients [14], and similarly, in 2007, a validation study that included 500 subjects reported 8.8% of the Latino population in Texas had melasma, while 4.0% reported a history of it [15]. In Ethiopia, the prevalence of melasma was reported as 1.5% in a study conducted between September 1995 and August 1996, involving 7760 patients with 9725 dermatological diseases [16]. More recently, in 2019, a cross-sectional study conducted in public hospitals in Durban, South Africa reported that dyschromia, including melasma, are the third most prevalent dermatologic diagnosis in Durban [17].

Although asymptomatic, melasma as a facial disorder affects the appearance of facial skin aesthetically and can reduce a person’s confidence, resulting in a low quality of life for the patient [7,8]. Hence, personal and socioeconomic factors have been shown to have an impact on health-related QoL [18]. QoL is defined as the ability to perform daily activities appropriate to a person’s age and plays a significant role in society [7,19]. The World Health Organization (WHO) describes quality of life as an individual’s perceptions of their position in life within their cultural context, value systems, expectations, goals, morals, and concerns [7,19,20,21].

Melasma is often associated with a variety of factors such as sun exposure, genetics, sex steroids (pregnancy and oral contraceptives), drugs, or cosmetics [7,8,22,23,24]. It is caused by melanocytic hypertrophy and a hyperfunction of the epidermal–melanic unit [25]. Some studies have revealed that the pathology of melasma points to a more heterogeneous pathogenesis, involving interactions between keratinocytes, mast cells, gene regulation abnormalities, neovascularization, and the disruption of the basement membrane [24,26,27]. Due to this complex pathogenesis, melasma is difficult to target and likely to recur after treatment. Oral therapies (tranexamic acid, glutathione), procedural interventions (chemical peels, microneedling, lasers, and lights), and topical therapies (tretinoin, hydroquinone, triple combination) are helpful; however, they are not suitable for all skin types due to undesired side effects and suboptimal results, especially when dealing with darker skin types (Fitzpatrick skin types IV–VI) [9,28].

The melasma quality of life scale (MELASQoL) is one of the validated dermatology-specific instruments used to assess the impact of melasma on health-related quality of life (HRQoL) and has been established through clinical studies and validated in several languages [20,29,30,31]. The MELASQoL questionnaire comprises a 10-item questionnaire, based on SKINDEX-16 [30] and is used in numerous countries [7,29,32,33,34]. However, there is no evidence that it has been fully explored in clinical practice in South Africa. With an emphasis on people with darker skin types, the aim of this study is to comprehensively understand melasma by achieving the following objectives:Assess the severity of melasma;Evaluate the impact of melasma on the QoL of affected patients;Identify predictors of QoL through stepwise regression analysis.

Although it may be possible for a clinician to obtain an overall view of a patient’s QoL by asking a single question, the use of a more detailed questionnaire provides much richer detail that allows the clinician to address both specific problems experienced by a patient and to identify which aspects of the patient’s life are most severely affected by their disease. Intervention can therefore be directed more appropriately [35].

The research questions that guided the study were as follows:What are the key factors influencing the severity of melasma in individuals with darker skin types (Fitzpatrick skin types IV–VI)?How does melasma impact the QoL of individuals with darker skin types?Can a predictive model be developed to assess the impact of melasma on the quality of life?

To the best of our knowledge, no studies have been conducted in South Africa on this important topic. We believe findings from this study will play a major role in informing dermatologists in their clinical decision-making on a routine basis. Hence, measuring QoL can help enhance patient care by identifying the need for supportive interventions and also help to track the improvement of patient HRQoL as well as influence healthcare policy.

## 2. Materials and Methods

### 2.1. Study Design and Setting

We conducted a cross-sectional study from three private dermatology clinics (Heritage House-Musgrave, Multimedics-Umhlanga, and Durdoc-Durban CBD) in Durban.

### 2.2. Study Population

We enrolled only patients who provided consent to participate in the study, who met all the inclusion criteria (Table 1).

### 2.3. Sample Size and Sampling Techniques

We enrolled a total of 150 patients from an existing database. The sample size was determined based on previous studies that have used similar methodologies to assess melasma severity and its impact on QoL [30,36]. Our current study mirrors Dlova et al.’s 2019 research, in which the authors sought to use their prevalence data to enhance dermatological services in KwaZulu-Natal and South Africa as a whole [17]. This alignment with their objectives led us to choose the same sampling location. The questionnaire was specifically designed with the research topic and objectives in mind, and it was piloted with a small group of participants (*n* = 5) to ensure its utmost clarity and relevance. The survey was administered to patients in English, either online or face-to-face by a trained multilingual interviewer. All the surveys that were filled in manually were uploaded by trained data capture personnel. Please see Appendix A for more information on the various sections of the questionnaire https://figshare.com/s/bba697794f459ca721c1 (accessed on 15 November 2023).

### 2.4. Data Collection

Data were collected between March and December of 2022. All included patients had a dermatologist-based diagnosis of melasma and a further detailed clinical examination by the same dermatologist. At enrolment, all patients filled out a clinical survey form to obtain information on their demographics: age, gender, marital status, family history, sites of involvement, as well as the use of cosmetics or other treatment alternatives for melasma. Melasma distribution was divided into three regions: centrofacial (cheeks, forehead, upper lip, nose, and chin); mandibular (ramus of the mandible); and malar (cheeks and nose). In addition, data on the disease chronicity, aetiological factors, including occupation, sun exposure, pregnancy history, the use of hormone replacement therapy (HRT) amongst postmenopausal women, oral contraceptives use, and any other associated conditions with onset of melasma, were collected. The MELASQoL questionnaire was administered to respondents to measure their QoL. MELASQoL has been shown to be valid and reliable measures of the impact of melasma on quality of life [30,34,37]. The components of the MELASQoL scale were added up to present a final score. The lowest score, i.e., 1 point for each 7 factor equals 7. Or if the respondent scored 10 for each factor, the MELASQOL would be 70 as per the developers of the MELASQOL instrument. The severity of melasma was graded based on MASI [25]. The higher the MASI scores, the more severe the melasma [25].

### 2.5. Statistical Analysis

Data were collected, processed, and analyzed using SPSS version 28 software. Descriptive statistics on continuous data were conducted whilst frequencies were reported for categorical variables. A significance level of 0.05 was employed. A Pearson correlation matrix was used to establish the presence of multicollinearity between predictor variables. Despite the application of stepwise regression being considered crude by most statisticians, it is still widely reported in the literature and remains an invaluable tool in evaluating predictors [38]. A stepwise regression was performed to establish statistically significant predictors of QoL [39]. The stepwise regression algorithm identifies predictor variables with *p*-values for the *F*-statistic ≤ 0.050, which are considered for inclusion in the model, while variables with *p*-values ≥ 0.100 are removed from the model. These criteria guide the automated selection and removal of predictor variables based on their statistical significance in relation to the dependent variable [40]. Before the stepwise regression, the variables were evaluated for the presence of multicollinearity. High correlation values between predictors lead to redundancy and may markedly influence the model’s predictive value. As a guideline, any correlation value with an absolute value greater than 0.700 should be removed from a regression model.

### 2.6. Ethical Consideration

The study was carried out according to ethical principles, and in accordance with the Declaration of Helsinki. Patients were provided with information about the nature of the study and were only recruited after informed consent was received. They were assured that their anonymity and confidentiality would be maintained. We ensured anonymity of the participants, and that the research does not cause harm to either participants or others. Furthermore, the study was approved by the University of KwaZulu-Natal Biomedical Research Ethics Committee (UKZN BREC) (Protocol reference number: BREC/00002721/2021). Informed consent was obtained from all participants. The survey was conducted between March and December 2022.

## 3. Results

### 3.1. Study Respondents’ Characteristics

This study enrolled a total of 150 respondents of either sex. The mean age of the respondents was 47.30 years (*SD* = 10.21). The majority (*n* = 143, 95%) were female, while the remaining were males (*n* = 7, 5%). One hundred and fourteen (76%) of the participants were black African, 13 (9%) were Indian and 23 (15%) were of mixed ancestry. Perceived causes of melasma included numerous triggers as illustrated in Figure 1 below. A few respondents did not know what caused their melasma.

Most respondents (61%) had no family history of melasma, while the remaining (39%) had a relative who suffered from melasma. Most people (41%) had their mother suffering from melasma, followed by their sibling (sister), aunt, cousin, and then lastly, uncle and brother. Some respondents (35%) had suffered from melasma for five years, while the majority of the respondents had experienced melasma in the past six months. Most respondents (61%) had used some form of traditional intervention, e.g., turmeric powder paste, red ochre soil, or bark paste to treat their melasma, while the remaining (39%) indicated that they had used only dermatological treatment interventions. The most common area (40%) of melasma was on the cheeks, followed by the forehead sides of the face, jawline, and nose, respectively.

### 3.2. Descriptive Analysis of the Dependent and Predictor Variables

The mean and standard deviations of the respondent’s responses are shown in Table 2 These statistics describe all the variables of the sample population. Twenty-eight predictor variables were assessed.

These included the number of children (Kids), number of skin products used (Makeup), their age (Age), the Melasma Severity Index (Masi), time spent in the sun per day (SunOften), the sun protection factor used (spf), time spent in the sun (SunExposure), how long they have suffered with melasma (HowLongSuffer), how long they have been treated for it (TreatLongTreat), the number of triggers (Triggers), how often they play outdoor sports (SportOften), their gender (Gender), level of education (Educ), whether they use sun protection (SunPrt), whether they were consulting a doctor (Doctor), whether they understood the meaning of the word “Melasma” (Word), whether they were diagnosed as suffering from the condition (Suffer), the location of the condition (Forehead, Cheeks, Jawline, Nose, Sides), whether they were receiving treatment (Treatment), whether they were using plant-based remedies (Plants), whether they had family members also suffering from the condition (Family), whether they played sports (Sport), whether they were menopausal (Menopause), and whether they were on hormone replacement therapy (HRT).

Table 3 illustrates the frequencies of these key categorical variables.

Of particular significance in both Table 2 and Table 3 is that the respondents have a relatively high MELASQoL score (*M* = 56.29, *SD* = 7.35), are mainly women (95%), have had children (*M* = 2.10, *SD* = 1.13), and are middle-aged (*M* = 47.30, *SD* = 10.21). However, only 10% reported that they are menopausal.

### 3.3. Severity of Melasma Index (MASI)

Based on 150 observations, a descriptive analysis was performed on the MASI index. The mean MASI index score was 40.62 (*SD* = 4.87), with scores ranging from 31.00 to 48.00. The kurtosis value of −1.03 indicated a modestly platykurtic distribution, indicating fewer outliers than a normal distribution. The skewness value of −0.18 indicated a minor leftward bias, indicating a slightly negative skewness. The analysis also computed a confidence level of 95.0% (CI = 0.786) for the estimated population mean based on the sample data (Table 4).

These findings provide a descriptive summary of the MASI index, highlighting the mean level, variability, and distribution of the scores. Figure 2 below also shows their respective frequency distributions (histograms).

### 3.4. QoL (MELASQoL)

Table 5 shows the 10 themes tested in the survey that constitute the respondents’ MELASQoL scores when summed together. For all ten themes, the frequency distributions are negatively skewed, i.e., long tails to the left. This implies that respondents indicated that they were affected by melasma irrespective of their underlying melasma conditions, severity, outdoor behaviors, or demographics. These distributions indicate that even when the severity of melasma was low (MASI), and irrespective of which ethnic group or gender, their quality of life (MELASQoL) was still affected by melasma.

### 3.5. Stepwise Regression Analysis

The MELASQoL score of respondents was predicted using a multivariate regression analysis that considered all the 28 independent variables relevant to the primary research question, i.e., predicting the MELASQoL score of a melasma patient. Before being used in regression analysis, the majority of the independent variables had to be recorded because they were categorical variables.

Categories were dummy-coded as “0” or “1”. Depending on whether they fall under a particular category or not, people were assigned a code of “0” or “1”. These categories were explicitly defined as mutually exclusive. For example, if a respondent did not use sun protection, this response would be coded “0”. If they did use sun protection, the response was coded “1”. The coding did not allow for any overlapping responses. The frequency of these dummy predictor variables is shown in Table 3.

The stepwise regression method was used to build the regression model in SPSS version 28. Starting with all 28 predictor variables in the study question, this method includes removing each variable one at a time. Four variables had a Pearson correlation coefficient with absolute values greater than 0.700. These variables were: sun exposure and the use of sun protection with a correlation coefficient of 0.781, and sport participation and outdoor sport participation with a correlation coefficient of 0.776. These variables were however not removed, but rather allowed the stepwise regression algorithm to include or exclude them objectively. The correlation matrix is reflected in Table 6.

Table 7 shows that the stepwise regression produced four statistically significant models and the increasing value of *R*^2^ and falling standard errors with a successive inclusion of the independent variables from Model 1 (with MASI only) to Model 4 (with predictors MASI, Cheeks, Education, and Menopausal). *R*^2^ improved from 0.044 in Model 1 to 0.145 in Model 4. Model 4 produced the highest *R*^2^ and an adjusted *R*^2^ of 0.145 and 0.122, respectively, with the lowest standard error of the estimate (*SE* = 6.889).

Notwithstanding this low *R*^2^ value, different authors, depending on inter alia the regression model and other factors such as the context of the study, have different opinions concerning the informational utility derived from the use of *R*^2^. Falk and Miller, 1992 [41] suggested that for the variance explained of a specific endogenous construct to be judged appropriate, *R*^2^ values should be equal to or more than 0.10. Cohen, 1988 [42], proposed the following *R*^2^ values for endogenous latent variables: 0.26 (substantial), 0.13 (moderate), and 0.02 (weak). Whilst Chin, 1998 [43], suggested *R*^2^ values of 0.67 (substantial), 0.33 (moderate), and 0.19 (weak) for endogenous latent variables. An *R*^2^ of 0.145 illustrates the dire need to research this topic more in the future to refine the explanatory power of future regression models.

Finally, the Durbin–Watson statistic is calculated to be 1.951, indicating the absence of autocorrelation. The ANOVA (Table 8) showed that all four models produced significance values less than 0.05, with the final model, Model 4, having an overall significance of *F* (4, 145) = 6.153 and *p* < 0.001.

When considering only Model 4, the final predictors are all statistically significant at a 0.05 level of significance, i.e., MASI (β = 0.209, *t* = 2.628, *p* < 0.001), Cheeks (β = −0.268, *t* = −3.405, *p* < 0.001), Education (β = −0.159, *t* = −2.029, *p* = 0.044), and Menopausal (β = −0.161, *t* = −2.027, *p* = 0.045). Moreover, the sign of their standardized β-values was also evaluated. The sign of the MASI coefficient was positive (β = +0.209). Intuitively, this understandable in that a higher MASI score, i.e., a high assessed severity of the respondent’s melasma should be positively correlated to a lower respondent’s QoL. The other three predictors have negative standardized β-values. The implication of this can be summarised as follows: Cheeks (β = −0.268), Education (β = −0.159), and Menopausal (β = −0.161). The Variance Inflation Factor (VIF) was also used to determine the presence of multicollinearity. Table 9, therefore, shows that the presence of multicollinearity is not a concern.

Finally, the prediction of the MELASQoL score is described by the following equation based on the stepwise Model 4 regression above:
$\hat{MELASQoL}$=51.730+0.315 Masi−4.686 Cheeks−4.148 Education−2.519 Menopausal
*se*=(5.249)
(0.120)
(1.376)
(2.044)
(1.243)*r*^2^ = 0.145*t*=(9.856)
(2.628)
(−3.405)
(−2.029)
(−2.027)*F* (4, 145)= 6.153*p*=(<0.001)
(0.010)
(<0.001)
(0.044)
(0.045)*p* < 0.001

## 4. Discussion

The study findings revealed that melasma has a severe impact on the QoL of patients with darker skin types who suffer from it. The results of the study indicated that patients who suffer from melasma often describe feelings of embarrassment, low self-esteem, anhedonia, and a lack of willingness to socialize; suicidal tendencies have also been reported. Through the stepwise regression model, we distilled four key predictor variables out of 28, and a regression model to predict MELASQoL was developed, given the four predictors: MASI (β = 0.209, *t* = 2.628, *p* < 0.001), Cheeks (β = −0.268, *t* = −3.405, *p* < 0.001), Education (β = −0.159, *t* = −2.029, *p* = 0.044), and Menopausal (β = −0.161, *t* = −2.027, *p* = 0.045). The equation’s significance enables the remote scoring of MELASQoL based on the four variables, which could help adapt therapeutic interventions depending on the predicted score.

The influence of melasma on patients’ QoL was reflected through both emotional distress and social life. Regarding their skin condition, respondents expressed dissatisfaction, despair, embarrassment, and depression. They revealed that it made them feel unattractive and that it had an impact on their social livelihoods. Melasma causes patients to feel unattractive to others and tends to decrease their desire to be around or interact with them. The reported epidemiologic characteristics of melasma patients in the current study were similar in some respects to those in previously reported factors [23,33,44,45,46].

### 4.1. The Use of Sunscreen

Our results indicate that 89.33% of respondents use some form of sunscreen in their skincare routine while 10.67% used no sunscreen. The few respondents who indicated never using sunscreen were mainly women. Figure 3 below provides an illustration of reasons for not using sun protection creams.

One of the main reasons provided by participants for not using a sunscreen was the affordability factor. Some participants cited that they experience skin reactions, irritation, or sensitivity to sun protection cream, which discourages them from using it. Some avoid using sun protection cream because they believe it makes their face look white or pale and “creates” breakouts, while some do not see the necessity of using sun protection cream, particularly if they spend most of their time indoors or do not spend much time in direct sunlight. A few participants report using moisturizers or other products that have an SPF as an alternative to dedicated sun protection cream, while some believe that sun protection cream is ineffective or does not work as advertised. Some participants simply responded with “N/A” or “none”, indicating that they have no particular reason for not using sun protection cream, while others did not provide a reason at all.

A large proportion of respondents (92.48%) use sunscreen regularly but only once a day (92.48%), which may not be sufficient protection. This once-off sun protection application could be attributed to limited knowledge about the proper application of the sunscreen and its ability to protect against photo-pigmentation and the value of the SPF. Many people believe that higher-SPF sunscreens provide adequate protection throughout the day [47]. Sunscreens must be applied in an amount of 2 mg/cm^2^ to provide the SPF stated on the container [48]. However, several studies have shown that consumers apply much less, only about a quarter (0.5 mg/cm^2^) of the recommended amount; therefore, the reapplication of sunscreen has been recommended to address these problems [48,49,50]. Furthermore, sweating, movement, and failure to reapply sunscreens at regular intervals all contribute to sunscreens performing poorly in the field when compared to their predicted efficacy in the laboratory [51,52]. It is important to disseminate the message that extreme caution is required in preventing the sun from aggravating melasma on the skin, which necessitates increased effort on the part of skin care specialists to educate and actively engage patients in effective sunscreen application.

Similar behaviors and attitudes concerning the use of sunscreens have been previously reported from respondents with colored skin [53,54]. Given the social–political background in South Africa, misconceptions about the use of sun protection still exist, and yet, the literature demonstrates that all skin types need to be protected from solar ultraviolet radiation (UVR) [55,56]. Due to its geographic location, South Africa is a very hot country with daytime ambient temperatures that often exceed 35 °C, the levels of ambient solar (UVR) throughout most of the year are high with the UV index (UVI) being frequently extreme (11+ or >6400 Jm^2^/day) [57,58]. In our study, four variables had Pearson correlation coefficients with absolute values greater than 0.700. These variables were: sun exposure and the use of sun protection with a correlation coefficient of 0.781, and sports participation and participation in outdoor sports with a correlation coefficient of 0.776. Also, a strong correlation between MASI and Sun Exposure, *R* = 0.663 (Table 6) was noted in our study. Jointly, studies provide evidence that excessive sun exposure contributes to melasma and therefore impacts the severity of melasma [59,60,61]. The excessive sun exposure from our study may have resulted from the participation in outdoor sports as well as insufficient sun protection application, as indicated by the study respondents. Hence, future interventions should incorporate components to effectively minimize barriers to sun protection and improve their self-efficacy in wearing sunscreen.

### 4.2. Genetic Predisposition, Aggravating Factors, and Product Use

Respondents reported a family history of at least a first- and second-degree relative suffering from melasma, suggesting a genetic predisposition as previously indicated in the literature [36,62]. Although men were the minority group (4.67%) in our study, they indicated similar effects of melasma as females. This finding is similar to previous reports that men are equally affected by melasma [46,62,63,64]. Aggravating and triggering factors (Figure 1) were similar for both male and female respondents. A few respondents (39.33%) indicated that they use alternative or homemade interventions such as “mmemezi” bark, lemon and/or turmeric powder paste, and clays. The potential use of alternative treatments in managing uneven skin tone is gaining popularity as these treatments are perceived as being safe, affordable, and easily accessible, and they provide protection from sun damage [28,65]. Most respondents (86%) reported that they opt for professional-based treatments such as chemical peels. Multiple product use included both over-the-counter and prescribed creams such as hydroquinone, retinol, and vitamins like vitamin C and vitamin A. A majority of the respondents mentioned that they use specific brands such as Garnier, Eucerin, Dermalogica, and La Roche-Posay products, which are known for skin lightening. They also listed multiple products or treatments they use, often including a combination of creams, serums, and sunscreen. Most of these creams contain glycolic acid, anti-oxidants, and vitamins C and E, which are common ingredients used for skin lightening [66,67,68].

### 4.3. The Quality of Life

MELASQoL was created from questions more relevant to melasma-specific HRQoL issues, with a focus on the emotional and psychological aspects [30]. When compared to the DLQI and SKINDEX 16, MELASQoL was found to have high internal consistency, validity, and discriminatory power [29,30]. There is strong evidence that melasma can strongly affect quality of life [5,7,20,31,34,44,69]. Respondents reported a relatively high MELASQoL score (*M* = 56.29, *SD* = 7.35), indicating a significant influence of melasma on patients’ quality of life. Other studies in different countries have reported means of 55.00 ± 10.60 (Australia) [70], 44.4 ± 14.19 (Brazil) [33], 44.40 ± 14.90 (Brazil) [20], 42.49 (Spain) [31], 39.97 ± 12.07 (Indonesia) [8], and 38.10 ± 16.60 (Republic of Korea) [71]. In all these studies, melasma is reported to cause frustration, embarrassment, and a loss of confidence among respondents; furthermore, it makes them feel unattractive and it affects their relationships.

Other studies which analyzed the relationship between the MASI and MELASQoL suggested that there is a statistically significant correlation between the two scores [7,29,31,32,71,72]. Hence, our study showed that MASI and MELASQoL scores were statistically correlated (*R* = 0.222) (Table 6). However, contrary to this popular view, some studies have shown an unrelated or weak correlation between MASI and MELASQoL [8,33,34,37,44,70,73]. Thus, the relationship between the MASI and MELASQoL scores is mixed. Clinical severity should not be the only criterion used to assess the burden of patients’ skin conditions. The MASI score is based on “feelings”, and since they change according to the situation, “feelings” lack a clear criterion of evaluation. Even when melasma is not severe, it can cause emotional stress, potentially reducing patients’ quality of life. Given the present study, it is evident that melasma has a significant negative impact on a patient’s QoL. The MELASQoL may not be an ideal tool to measure QoL as it mainly focuses on emotions, which makes the measurement subjective. Previously, a more objective instrument that examines the severity of melasma has been proposed. The new modified MASI (mMASI) score is based on the measurement of darkness and area of involvement in identifying melasma severity, and since homogeneity is unreliable, it has been removed from the new modified MASI score [74]. In assessing the severity of melasma, most authors have agreed that the mMASI score is reliable, accurate, and responsive to change. Furthermore, the mMASI score has been demonstrated to be easier to acquire and perform, as well as simpler to calculate, than the MASI score [75,76,77]. Thus, the mMASI score can successfully substitute the MASI score.

Our study results showed that the MELASQoL score is impacted, so it was decided to not remove the variables and allow the stepwise regression algorithm to include or exclude them objectively. This decision was supported by the model’s collinearity statistics in Table 9. Myers [78] suggests that a tolerance value below 0.1 indicates a serious collinearity problem, whilst Menard [79] recommends that a tolerance value less than 0.2 indicates a potential collinearity problem. Once again, as a rule of thumb, a tolerance of 0.1 or less is a cause for concern. Similarly, the Variance Inflation Factor (VIF) is also used for determining the presence of multicollinearity. Values of VIF exceeding 10 are often regarded as indicating multicollinearity [80]. The rule of thumb is that VIF must be less than 5.0. As depicted in Table 9, the presence of multicollinearity is not a concern.

### 4.4. The MELASQoL Predictors

The fourth model in the stepwise iteration presented MASI (β = 0.209, *t* = 2.628, *p* < 0.001), Cheeks (β = −0.268, *t* = −3.405, *p* < 0.001), Education (β = −0.159, *t* = −2.029, *p* = 0.044), and Menopause (β = −0.161, *t* = −2.027, *p* = 0.045) as statistically significant predictors of MELASQoL at a 0.05 level of significance with negative standardized β-values. The *R*^2^ value was 0.145, implying that 14.5% of the changes in MELASQoL can be accounted for by these four independent variables. The overall model was found to be statistically significant with *F* (4, 145) = 6.153 and *p* < 0.001. During the stepwise regression, it was noted that a lot of predictors were dichotomous, and because these binary variables are typically mutually exclusive, it is therefore standard practice to code them as 0 or 1 [81,82,83]. Furthermore, it is good practice to check for multicollinearity so that the final model is parsimonious; hence, a multicollinearity test was performed [84,85,86]. The implication of this can be summarized as follows.

The value for Cheeks (β = −0.268) implied that the greater the prevalence of melasma on the malar area of the respondent, the lower their reported MELASQoL, i.e., the higher their reported QoL. Previous research has identified the malar to be one of the most common patterns of melasma presentation [5,69,87,88,89,90]. In our study, we found that people who had a malar pattern of melasma reported that they were not negatively affected by melasma. Some studies have reported that the progression of the disease has no bearing on the quality of life [37,44,70]. Another reason could be that it is easier to cover melasma in the malar area with cosmetic camouflage [91,92,93].

The value for Education (β = −0.159) implied that the more educated the respondent, the higher the reported MELASQoL score, i.e., a lower QoL. This finding could be due to the fact that, in socially unequal societies like South Africa, educated people are more prominent in society, and therefore, meeting people in their qualified professions may make them feel negatively affected by their condition. Concerning the level of education, some studies have shown that people with a low level of education may have less information regarding disease prevention, are more likely to work in less qualified fields of work, be more vulnerable to unprotected sun exposure, may have less access to dermatologic care, and may not be able to afford costly treatments and make-up [31,45,72,94,95].

With regard to the value for Menopause (β = −0.161), when women are menopausal, the lower the MELASQoL score, the greater their overall quality of life. This finding may be attributed to a few suggestions. Firstly, it may be that due to their age; they have accepted the condition of their skin; hence, they have suffered for many years and therefore are no longer bothered. Another reason could be that they might have slowed down in pursuing careers where they have to meet new people. The lower MELASQoL score on menopausal women may be attributed to a prior knowledge of the implications of their hormone production stage, as menopause is known to aggravate the severity of melasma. Women who are menopausal produce significant amounts of estrogen, which is a known risk factor for melasma [31,96,97,98].

## 5. Future Perspective and Significance of the Study

The impact of melasma on QoL is extensive and requires an inclusive response. Melasma is difficult to treat and is prone to recur after treatment due to its multifaceted etiology. Although oral medicines, procedural interventions, and topical treatments are beneficial, they are not appropriate for all skin types due to undesirable side effects and inadequate results, particularly for individuals with darker skin types. Future research on genetic factors impacting susceptibility to melasma could contribute to our understanding of its complex etiology, given the multifaceted character of the condition.

Although asymptomatic, melasma is reported to have a negative impact on those who suffer from it. Dealing with melasma in a holistic way, including the emotional aspect, is crucial. Lessons gleaned from this study indicate that melasma is not merely a cosmetic concern, but rather a medical issue, as it affects the QoL. Arguably, this lesson emphasizes the importance of innovative and adaptive individualistic treatment approach when dealing with melasma patients. Looking beyond the physical components, future studies should examine how mental health therapies affect melasma outcomes, which could improve the patient-centered care model. For example, the prediction of the MELASQoL score as described by the equation based on the stepwise Model 4 regression can allow for the remote scoring of MELASQoL based on the four variables (MASI, Cheeks, Education, Menopause), which could help customize the treatment intervention and consequently possibly help improve patient outcomes.

The remote scoring model is arguably more robust and parsimonious. Overall, the outcomes of this study provide useful insights into improving tailored and comprehensive therapy protocols when dealing with melasma patients. Building on remote scoring models, future studies could explore the efficacy of novel topical treatments, potentially introducing safer and more effective options for managing melasma. In addition to the holistic treatment plan, people with darker skin types should be informed about the correct application of sun protection creams and be encouraged to use broad-spectrum photo-protection covering the solar spectrum from UVB to blue light.

In summary, this study conclusively suggests that by adopting and implementing a holistic approach when treating melasma, the QoL will be improved, thereby promoting better mental health outcomes. Thus, it will catalyze the promotion of the sustainable development goal 3 (SDG3), which involves providing health and wellness to all. Future studies may examine the relative efficacy of different treatment techniques across multiple skin types, acknowledging the difficulties in treating darker skin types and assuring more inclusive and individualized approaches.

## 6. Limitations of the Study

Notwithstanding the fact that the data may not necessarily be representative of the complete health care population of the study area, the authors acknowledge this limitation. This region, however, does represent the most populous region for darker-skin-type women in South Africa, who are steeped in cultural norms and practices, thereby providing a good test case for future research. The selected participants’ experiences with melasma, may not fully represent the diversity of views and the QoL in the Durban community as a whole as only darker-skin-type people were enrolled in the study. Additionally, the small number of men enrolled in this study is not a true representation of the population. Also, people who reside in the rural areas could express different results due to dissimilar lifestyle habits. To further address potential differences in healthcare outcomes and access, future studies might investigate the impact of socioeconomic factors on the availability of melasma treatment. This would add to the ongoing conversation about constraints.

## 7. Conclusions

Melasma has a significant impact on a patient’s quality of life (QoL). In this study, we found that an impairment on the quality of life is greater irrespective of the underlying melasma conditions. Even when melasma is not severe, it can cause emotional stress, potentially reducing patients’ quality of life. Through the stepwise regression model, we distilled four key predictor variables out of 28 and developed a regression model to predict MELASQoL, given these four predictors. The significance of the equation can allow, for example, the remote scoring of MELASQoL based on the four variables, which could help customize the treatment intervention based on the forecasted score. The authors acknowledge that the study may not comprehensively represent the entire healthcare population, this region stands as the most densely populated area for women with darker skin types in South Africa, potentially experiencing a high incidence of dermatological issues. As such, it serves as a valuable test case for future research. The study concludes with suggestions for future research that compare the experiences of melasma in urban and rural settings, considering the differences in lifestyle behaviors that may affect treatment outcomes and ultimately enhance a patient’s quality of life.

## Figures and Tables

**Figure 1 ijerph-20-07068-f001:**
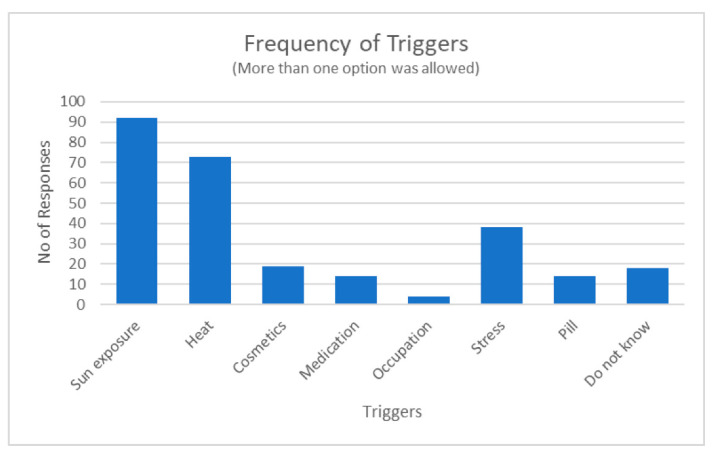
Number of times a trigger category was selected by the respondents.

**Figure 2 ijerph-20-07068-f002:**
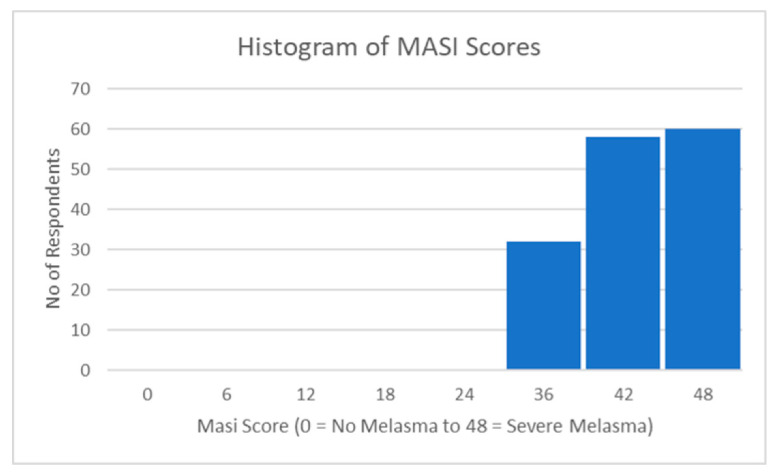
Distribution of MASI scores.

**Figure 3 ijerph-20-07068-f003:**
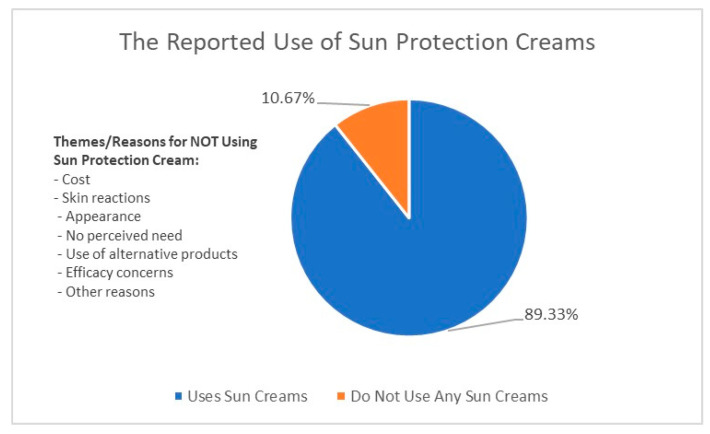
Reasons for the nonuse of sun protection.

**Table 1 ijerph-20-07068-t001:** Inclusion and exclusion criteria.

Inclusion	Exclusion
patients older than 18 years with existing facial melasma	vulnerable people and minors
male and female patients with Fitzpatrick skin types IV–VI	Fitzpatrick skin types I–III
black Africans, Indians and mixed ancestry people	lighter skin types
all types of melasma: epidermal, dermal or mixed facial melasma	other hypermelanosis skin disorders

**Table 2 ijerph-20-07068-t002:** Descriptive statistics of the dependent and predictor variables.

Variables	Notes	Mean	Std. Deviation
MELASQoL	Scale from 7 to 70	56.29	7.35
Children	Number of children	2.10	1.13
Skin care regime	Total number of skin products used	2.43	2.11
Age	Years	47.30	10.21
MASI	MASI grading scale	40.62	4.87
Sun exposure	Times a day	0.95	0.42
SPF	Range from 4 to 100	45.09	27.64
Sun exposure	Minutes per day	113.00	47.33
Duration of melasma	Years	6.38	4.78
Treatment duration	Months	10.54	13.56
Triggers	Total number of triggers	1.81	0.72
Sport participation	Number of days per week	1.01	2.22

**Table 3 ijerph-20-07068-t003:** Frequency table of dummy regressor variables.

Dummy Variables	Frequency of “0”	Percentage (%)	Frequency of “1”	Percentage (%)
Gender	7	5%	143	95%
Education	13	9%	137	91%
Use of sun protection	16	11%	134	89%
Previously consulted with a doctor	20	13%	130	87%
Familiarity with the word melasma	16	11%	134	89%
Suffers from melasma	9	6%	141	94%
Forehead	88	59%	62	41%
Cheeks	34	23%	116	77%
Jawline	116	77%	34	23%
Nose	134	89%	16	11%
Sides of the face	111	74%	39	26%
Current melasma treatment	18	12%	132	88%
Use of plants as an alternative treatment	91	61%	59	39%
Family history	91	61%	59	39%
Participation in outdoor sports	117	78%	33	22%
Post-menopausal	101	67%	49	33%
HRT	140	93%	10	7%

Gender: 0 = Male, 1 = Female; Education: 0 = Uneducated, 1 = Educated; for the rest of the variables: 0 = No, 1 = Yes.

**Table 4 ijerph-20-07068-t004:** Descriptive summary of the severity of melasma observed in the respondents.

MASI Statistics	
Mean	40.62
Standard Error	0.40
Median	40.00
Mode	40.00
Standard Deviation	4.87
Sample Variance	23.73
Kurtosis	−1.03
Skewness	−0.18
Range	17.00
Minimum	31.00
Maximum	48.00
Sample size	150
Confidence Level (95.0%)	0.786

**Table 5 ijerph-20-07068-t005:** Percentage of answers for each MELASQoL question from melasma patients (*N* = 150).

MELASQoL	Likert Scale ^a^							Descriptives			Frequency Distribution
Questions on…	1	2	3	4	5	6	7	Count	Mean	Std. Dev.	
Appearance	0	0	0	3	38	77	32	150	5.920	0.735	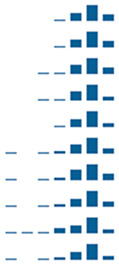
Frustration	0	0	0	4	37	77	32	150	5.913	0.748	
Embarrassment	0	0	2	7	40	70	31	150	5.807	0.862	
Depression	0	0	1	5	44	79	21	150	5.760	0.754	
Others	0	0	0	7	39	82	22	150	5.793	0.742	
Desire	1	0	1	13	40	71	24	150	5.667	0.943	
Affection	1	0	1	13	44	66	25	150	5.647	0.953	
Unattractiveness	1	0	2	10	39	74	24	150	5.693	0.938	
Unproductive	1	1	1	21	37	70	19	150	5.520	1.018	
Freedom	1	0	1	11	41	76	20	150	5.660	0.901	

^a^ 7-point Likert scale ranging from 1 = “Not bothered at all” to 7 = “Constantly bothered”.

**Table 6 ijerph-20-07068-t006:** Correlation matrix of predictor and dependent variables.

Variables		1	2	3	4	5	6	7	8	9	10	11	12	13	14	
1	Children	1.000														
2	Makeup	0.038	1.000													
3	Age	0.223	−0.040	1.000												
4	Masi	−0.057	−0.156	−0.140	1.000											
5	Sun Often	0.038	0.203	−0.070	−0.077	1.000										
6	SPF	−0.007	−0.045	0.295	−0.105	0.515	1.000									
7	Sun Exposure	−0.047	−0.175	0.035	0.663	−0.026	0.067	1.000								
8	How Long Suffer	−0.014	−0.013	0.328	−0.149	0.092	0.197	−0.035	1.000							
9	How Long Treat	0.021	0.439	0.139	−0.016	0.160	0.091	−0.035	0.153	1.000						
10	Triggers	0.023	0.213	−0.240	0.007	−0.095	−0.327	−0.104	−0.123	0.073	1.000					
11	Sport Often	−0.003	−0.024	0.134	0.009	0.036	0.130	0.119	0.264	0.000	0.102	1.000				
12	Gender	−0.008	0.076	0.170	−0.024	0.125	0.167	−0.013	0.174	0.046	−0.190	−0.028	1.000			
13	Education	−0.099	0.165	−0.114	0.092	0.191	0.161	−0.031	−0.090	0.135	−0.047	−0.095	−0.068	1.000		
14	Sun Protect	−0.008	0.205	−0.053	−0.038	0.781	0.562	0.031	0.091	0.126	−0.120	−0.018	0.128	0.201	1.000	
15	Consult Doctor	−0.052	−0.302	0.031	0.082	−0.043	0.193	0.042	−0.154	−0.051	0.007	0.028	−0.087	0.019	−0.008	
16	Know Word	0.108	−0.032	−0.117	0.012	0.013	0.021	−0.024	−0.042	0.054	0.212	0.021	−0.076	0.201	0.090	
17	You Suffer	0.047	−0.068	−0.192	0.121	0.039	0.052	0.123	−0.180	0.041	0.091	−0.050	−0.056	0.222	0.095	
18	Forehead	−0.014	0.097	−0.018	−0.037	−0.068	−0.062	−0.160	−0.022	0.085	−0.065	−0.076	−0.199	0.210	−0.061	
19	Cheeks	0.133	−0.055	−0.039	0.075	0.053	−0.019	0.061	0.008	0.076	0.015	−0.034	0.258	−0.167	−0.032	
20	Jawline	−0.176	−0.104	−0.039	0.076	−0.016	−0.063	−0.001	0.152	−0.065	0.008	0.185	−0.031	−0.060	−0.071	
21	Nose	0.046	0.103	0.011	0.066	−0.064	−0.095	0.024	0.070	0.128	0.090	0.136	−0.128	0.030	−0.090	
22	Sides Of Face	0.110	0.131	0.068	−0.046	0.102	0.153	−0.067	0.177	0.081	0.048	0.094	−0.013	0.129	0.156	
23	Use Treatment	0.106	−0.246	0.061	0.100	0.008	0.277	0.050	−0.074	0.270	0.018	0.029	−0.082	0.032	0.072	
24	Use Plants	−0.023	0.087	−0.135	0.023	0.024	−0.069	−0.071	0.078	0.045	−0.019	−0.169	−0.081	0.005	0.146	
25	Any Family	0.098	0.230	−0.135	0.026	0.154	−0.164	0.041	0.048	0.075	0.191	0.059	−0.016	0.151	0.146	
26	Outdoor Sport	−0.019	−0.002	0.099	0.013	−0.018	0.133	0.089	0.236	0.057	0.049	0.776	−0.035	−0.065	−0.077	
27	Menopausal	0.279	0.073	0.639	−0.227	0.010	0.191	−0.132	0.215	0.097	−0.176	−0.028	0.154	0.063	0.010	
28	HRT	−0.095	0.136	0.189	0.004	−0.034	−0.006	0.006	0.124	0.023	−0.117	0.096	0.059	−0.013	0.006	
29	MELASQoL	−0.158	−0.097	−0.086	0.211	−0.085	−0.140	0.113	−0.116	−0.157	0.117	−0.012	−0.096	−0.106	−0.100	
**Variables**		**15**	**16**	**17**	**18**	**19**	**20**	**21**	**22**	**23**	**24**	**25**	**26**	**27**	**28**	**29**
15	Consult Doctor	1.000														
16	Know Word	0.182	1.000													
17	You Suffer	0.066	0.640	1.000												
18	Forehead	−0.029	−0.017	0.098	1.000											
19	Cheeks	−0.025	0.071	0.064	−0.225	1.000										
20	Jawline	0.025	−0.071	−0.064	−0.066	0.027	1.000									
21	Nose	−0.055	−0.021	−0.004	0.236	0.084	0.380	1.000								
22	Sides of Face	−0.036	0.057	0.022	0.089	−0.224	0.042	0.189	1.000							
23	Use Treatment	0.519	0.271	0.166	0.060	0.094	−0.094	−0.072	0.032	1.000						
24	Use Plants	−0.046	0.057	−0.084	−0.094	0.012	0.086	−0.146	0.021	0.045	1.000					
25	Any Family	−0.166	0.101	0.088	−0.094	−0.020	0.118	−0.013	0.052	−0.165	0.190	1.000				
26	Outdoor Sport	0.019	−0.129	−0.205	−0.021	0.057	0.135	0.077	−0.021	0.048	−0.131	0.034	1.000			
27	Menopausal	−0.061	−0.036	−0.063	0.079	−0.132	−0.072	−0.056	0.106	−0.005	−0.095	−0.095	−0.027	1.000		
28	HRT	0.026	−0.081	−0.158	−0.116	−0.111	0.111	0.081	−0.158	−0.148	−0.051	0.004	0.181	0.099	1.000	
29	MELASQoL	−0.048	−0.101	−0.052	−0.121	−0.204	0.026	0.031	−0.019	−0.068	0.083	0.087	−0.025	−0.183	0.060	1.000

**Table 7 ijerph-20-07068-t007:** Stepwise regression model summary.

Model	*R*	*R* ^2^	Adjusted *R*^2^	Std. Error	Durbin–Watson
1	0.211 ^a^	0.044	0.038	7.210	
2	0.305 ^b^	0.093	0.081	7.047	
3	0.348 ^c^	0.121	0.103	6.962	
4	0.381 ^d^	0.145	0.122	6.889	1.951

^a^ Predictors: (Constant), MASI. ^b^ Predictors: (Constant), MASI, Cheeks. ^c^ Predictors: (Constant), MASI, Cheeks, Education. ^d^ Predictors: (Constant), MASI, Cheeks, Education, menopausal.

**Table 8 ijerph-20-07068-t008:** Analysis of the variance (ANOVA) of the four stepwise regression models.

Model ^a^		Sum of Squares	*Df*	Mean Square	*F*	Sig.
1	Regression	357.630	1	357.630	6.880	0.010 ^b^
	Residual	7692.768	148	51.978		
	Total	8050.398	149			
2	Regression	749.586	2	374.793	7.546	<0.001 ^c^
	Residual	7300.812	147	49.665		
	Total	8050.398	149			
3	Regression	973.157	3	324.386	6.692	<0.001 ^d^
	Residual	7077.241	146	48.474		
	Total	8050.398	149			
4	Regression	1168.115	4	292.029	6.153	<0.001 ^e^
	Residual	6882.283	145	47.464		
	Total	8050.398	149			

^a^ Dependent Variable: MELASQoL. ^b^ Predictors: (Constant), MASI. ^c^ Predictors: (Constant), MASI, Cheeks. ^d^ Predictors: (Constant), MASI, Cheeks, Education. ^e^ Predictors: (Constant), MASI, Cheeks, Education, Menopausal.

**Table 9 ijerph-20-07068-t009:** Regression coefficients.

Model		Unstandardized Coefficients	Std. Error	Standardized Coefficients	*t*-Values	Significance	Collinearity Statistics	
		B		Beta			Tolerance	VIF
1	(Constant)	43.376	4.960		8.745	<0.001		
	MASI	0.318	0.121	0.211	2.623	0.010	1.000	1.000
2	(Constant)	45.356	4.900		9.257	<0.001		
	MASI	0.343	0.119	0.227	2.886	0.004	0.994	1.006
	Cheeks	−3.872	1.378	−0.221	−2.809	0.006	0.994	1.006
3	(Constant)	48.710	5.086		9.577	<0.001		
	MASI	0.370	0.118	0.245	3.132	0.002	0.983	1.017
	Cheeks	−4.391	1.383	−0.251	−3.175	0.002	0.964	1.037
	Education	−4.426	2.061	−0.170	−2.148	0.033	0.961	1.040
4	(Constant)	51.730	5.249		9.856	<0.001		
	MASI	0.315	0.120	0.209	2.628	0.010	0.933	1.071
	Cheeks	−4.686	1.376	−0.268	−3.405	<0.001	0.953	1.049
	Education	−4.148	2.044	−0.159	−2.029	0.044	0.957	1.045
	Menopausal	−2.519	1.243	−0.161	−2.027	0.045	0.931	1.074

## Data Availability

Due to privacy, data are available upon request from the first author.

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
