# Peer review of "The Effect of Melasma on the Quality of Life in People with Darker Skin Types Living in Durban, South Africa"

_ijerph, 2023, doi:10.3390/ijerph20227068_

Round 1

Reviewer 1 Report

Comments and Suggestions for Authors

Comments:

Abstract:

The abstract is generally well-structured and provides a clear overview of the study's purpose, methodology, and key findings. However, there is still need to improve in following context.

  1. The p-values should be presented with the exact number of decimal places for precision (e.g., p < 0.001). The phrase "p < .010" should be presented as "p < 0.010" for clarity. c.
  2. The significance of the equation mentioned in the last sentence is not explained. It would be beneficial to briefly describe what the equation signifies.
  3. The abstract lacks specific information about the demographics of the study population, which could be valuable for readers to understand the context.

Introduction:

  1. The introduction starts well by explaining the significance of skin and skin conditions, but it could benefit from a more engaging and concise opening statement to capture the reader's attention.
  2. The prevalence rates cited in different regions should include the year or time frame when these studies were conducted for context.
  3. The introduction should transition more smoothly from discussing the epidemiology of melasma to the impact on quality of life.
  4. The introduction lacks to present a clear research hypothesis or research question

Materials and Methods:

Include the STROBE check list (as an Appendix), and verify if they are complying with all the items on the STROBE check list.

  1. There is a repetition of the study design, setting, and study population (line 99 to 106) information. This redundancy should be eliminated.
  2. The sample size should be justified or explained more thoroughly. Why was a sample of 150 chosen? What is the basis for this number?

3.      Explain if the questionnaire was pilot tested. Have the participants completed an informed consent? It remains to include the study population, the sample and the sampling.

4.      What were the inclusion and exclusion criteria?

  1. The data collection process should be described more concisely without repetitive information.
  2. While mentioning the use of a clinical survey form and examinations by a dermatologist is important, it would be beneficial to provide more details on the content of the clinical survey form and the criteria used for dermatological examination.
  3. The explanation of the variables in the stepwise regression model is limited. How were these variables selected, and what is the rationale for their inclusion?
  4. Specific details about the reliability and validity of the MELASQoL instrument used for data collection would be valuable.
  5. The significance level for the statistical analysis should be clearly stated in the methodology section.

Discussion:

  1. The discussion section effectively summarizes the findings but should begin with a concise recap of the study's main objectives and major findings.
  2. The discussion of respondents' sunscreen use and the reasons behind it is valuable, but it would benefit from a more structured presentation.
  3. The discussion about the influence of hormone production stages on melasma could be better integrated with the rest of the findings, rather than appearing as an isolated paragraph.
  4. The interpretation of the findings regarding education level and its impact on MELASQoL is somewhat speculative. It would be stronger with more concrete evidence and references to support the claims.
  5. The use of the mMASI score as a possible substitute for MASI is interesting, but it requires more context and a clear rationale for its use.

Conclusions:

6.      The data is not representative. As the data is not representative of complete health care population of Study area (Durban, South Africa). The authors should improve these conclusions on the basis of more robust analyses.

7.      Authors should describe any limitations of their study. Furthermore, these limitations should be discussed.

8.      Authors should add Future perspective and significance of the study.

9.      Authors should add abbreviation list in the manuscript.

Grammatical Errors/mistakes:

Authors should address identified minor grammatical issues to improve the clarity, readability and quality of the article. Some of the errors are listed below.

1.                 1.   And fourth most" should be "the fourth most."

  1. "Amongst" should be "among."
  2. "More especially" should be "especially."
  3. The phrase "quality of life for the patient" should be rephrased as "the patient's quality of life."
  4. In the selected skin clinics" should be rephrased to specify the criteria for clinic selection, as "selected skin clinics" is vague.
  5. As the developers of the MELASQOL instrument" should be rewritten for clarity, such as "as described by the developers of the MELASQoL instrument."

Comments on the Quality of English Language

Authors should address identified minor grammatical issues to improve the clarity, readability and quality of the article

Author Response

Attached please find the responses to the reviewer comments

Reviewer 2 Report

Comments and Suggestions for Authors

lines 129-133: please remove the underlined. 

lines 153-154: are these percentual right? 95%+8%=103%...I think the male are 5%...

Why the authors defined the Pearson correlation value 0.7 as the value to discussion. Which value from the correlation means p<0.05. For example, menopause and age has an interesting correlation. Also use treatment and Doctor. These findings can be interesting to be discussed, even with a lower correlation factor. I saw the discussion in the lines 258-267 but more than just discussion the R² values, the p-values seems to be also important. 

By the way, the paper is very well presented. 

347-349: did the male participants reported some kind of sexist barrier to start to use cosmetics and/or to look for treatment? If yes, please include this in your discussion. See reference below:

Infante, V. H. P., Bagatin, E., & Maia Campos, P. M. (2021). Skin photoaging in young men: A clinical study by skin imaging techniques. International journal of cosmetic science43(3), 341-351.

McKenzie, C., Rademaker, A. W., & Kundu, R. V. (2019). Masculine norms and sunscreen use among adult men in the United States: A cross-sectional study. Journal of the American Academy of Dermatology81(1), 243-244.

357-362: this finding could be a section in the results and the discussion should be explored. How the socioeconomic factors can affect it?

Author Response

Attached please find the response to the reviewer comments.

Thank you

Round 2

Reviewer 1 Report

Comments and Suggestions for Authors

Authors have diligently addressed almost all the comments and concerns raised during the review process. The revisions made have significantly improved the quality and clarity of the article. However, there are still need to improve the Future perspective and conclusion section. It would be more appropriate to concise the future perspective and conclusion based upon key findings. Furthermore, I recommend to add graphical presentation of inclusion & exclusion criteria of participants.

Author Response

The reviewer comments and responses are attached.

Thank you. 

Reviewer 2 Report

Comments and Suggestions for Authors

The corrections are fine, congratulations for your work.

Author Response

Reviewer comments and responses are attached below.

Thankyou
